# A Deep Learning Model for Classification of Endoscopic Gastroesophageal Reflux Disease

**DOI:** 10.3390/ijerph18052428

**Published:** 2021-03-02

**Authors:** Chi-Chih Wang, Yu-Ching Chiu, Wei-Liang Chen, Tzu-Wei Yang, Ming-Chang Tsai, Ming-Hseng Tseng

**Affiliations:** 1Institute of Medicine, Chung Shan Medical University, Taichung 402, Taiwan; bananaudwang@gmail.com (C.-C.W.); joviyoung@gmail.com (T.-W.Y.); 2School of Medicine, Chung Shan Medical University, Taichung 402, Taiwan; 3Division of Gastroenterology and Hepatology, Department of Internal Medicine, Chung Shan Medical University Hospital, Taichung 402, Taiwan; grincia@yahoo.com.tw; 4Master Program in Medical Informatics, Chung Shan Medical University, Taichung 402, Taiwan; cru912@gmail.com; 5Department of Medical Informatics, Chung Shan Medical University, Taichung 402, Taiwan; 6Information Technology Office, Chung Shan Medical University Hospital, Taichung 402, Taiwan

**Keywords:** gastroesophageal reflux disease classification, artificial intelligence, deep learning, conventional endoscopy, narrow-band image

## Abstract

Gastroesophageal reflux disease (GERD) is a common disease with high prevalence, and its endoscopic severity can be evaluated using the Los Angeles classification (LA grade). This paper proposes a deep learning model (i.e., GERD-VGGNet) that employs convolutional neural networks for automatic classification and interpretation of routine GERD LA grade. The proposed model employs a data augmentation technique, a two-stage no-freezing fine-tuning policy, and an early stopping criterion. As a result, the proposed model exhibits high generalizability. A dataset of images from 464 patients was used for model training and validation. An additional 32 patients served as a test set to evaluate the accuracy of both the model and our trainees. Experimental results demonstrate that the best model for the development set exhibited an overall accuracy of 99.2% (grade A–B), 100% (grade C–D), and 100% (normal group) using narrow-band image (NBI) endoscopy. On the test set, the proposed model resulted in an accuracy of 87.9%, which was significantly higher than the results of the trainees (75.0% and 65.6%). The proposed GERD-VGGNet model can assist automatic classification of GERD in conventional and NBI environments and thereby increase the accuracy of interpretation of the results by inexperienced endoscopists.

## 1. Introduction

Gastroesophageal reflux disease (GERD), which is a condition that develops when the reflux of stomach contents causes symptoms of discomfort and/or associated complications [1], is among the diseases with the highest prevalence over the past two decades [2,3]. GERD can be classified as either erosive or non-erosive esophagitis and is characterized by endoscopically visible breaks in the distal esophageal mucosa in the former category and a lack of such breaks in the latter [4,5]. Although double-contrast barium swallow examination has been previously used to diagnose GERD [6], esophagogastroduodenoscopy (EGD) is now the gold standard test for suspected GERD to evaluate the alarming features and/or the possibility of Barrett’s esophagus in high-risk patients [7,8]. In addition, the popular and powerful Los Angeles classification (LA grade) system, which was established more than 20 years ago, is used in endoscopy examinations to classify GERD [9].

Previous studies have focused on inter-observer [10] and intra-observer [11] variations of the LA grade, and those results found the agreement on LA grade between experienced endoscopists to be better than that of less experienced endoscopists [11,12]. Compared to conventional endoscopy, which is referred to as white light endoscopy, narrow-band image (NBI), which filters out the red spectrum of light, improves the consistency of esophagitis grading in both inter-observer and intra-observer settings in the previous study [13].

Due to technological improvements in artificial intelligence (AI), basic screening surveillance methods, e.g., chest X-rays, are likely to be replaced by AI in the near future. Deep learning comprises a series of computational methods that allows an algorithm to program itself by learning from a large number of examples that demonstrate the desired behavior without the need to regulate the rules. Compared to traditional machine learning algorithms that must extract image features based on manual experience, deep learning is a hierarchical feature learning architecture that can automatically capture features from image data for disease prediction. There are successful examples of AI applications in the medical field, e.g., radiology image interpretation [14,15], obstructive pulmonary disease recognition in computed tomography [16], diabetic retinopathy screening [17,18,19], esophageal cancer endoscopic diagnosis [20], dysplasia in Barrett’s esophagus, and detection of early gastric cancers [21]. Therefore, AI can assist or even replace basic interpretation techniques in the medical field.

To the best of our knowledge, only two studies have investigated GERD prediction using AI technologies. One study suggested that the combination between the QUestionario Italiano Diagnostico (QUID) questionnaire and an artificial neural network (ANN)-assisted algorithm is useful to differentiate GERD patients from healthy individuals but fails to further discriminate erosive from non-erosive patients [22]. The other study proposed a hierarchical heterogeneous descriptor fusion support vector machine (HHDF-SVM) method for GERD diagnosis from conventional endoscopic images [23]. However, the AI systems are only applicable to GERD prediction in binary classification. Therefore, to the best of our knowledge, our study is the first to develop a deep learning model for computer-aided diagnosis that focuses on automatic grading of GERD according to LA grades.

We attempted to train AI to identify GERD endoscopic features using the LA classification by developing a deep learning model from endoscopic images of the esophago-cardiac junction (EC-J). We then compared the performance of the endoscopic images under conventional and NBI endoscopy by the proposed AI model. We further evaluated the accuracy of the AI predictions and the results from trainees in an endoscopy society.

The remainder of this paper is organized as follows: the materials and methods are introduced in the Section 2. In the Section 3, the results and analysis of why improved results were obtained using the proposed method are explained in detail. The data results and results from other studies are discussed in the Section 4. The Section 5 is the conclusion.

## 2. Materials and Methods

For AI training development, endoscopic pictures of the EG-J were obtained retrospectively from the endoscopic system at Chung Shan Medical University Hospital. The quality of the endoscopic images and GERD classification were confirmed by two instructors at the Digestive Endoscopy Society of Taiwan. The images were taken from the records of 496 people who had received an EGD exam for either symptomatic diseases or as a health examination between December 2019 and March 2020. All images were deidentified prior to transfer to the study’s investigators, and all methods were performed according to relevant local regulations under the surveillance of the Institutional Review Board of Chung Shan Medical University Hospital.

### 2.1. Grading

We found that all endoscopic images were adequate for reviewing the entire structure of the EC-J. Notably, the brightness and contrast of the images were not artificially altered. The LA grading system was employed as the evaluation scale. An LA grade of A is described as mucosal breaks no longer than 5 mm that do not extend between the tops of two mucosal folds, a grade B includes mucosal breaks of more than 5 mm in length that do not extend between the tops of two mucosal folds; and a C–D grade includes one (or more) mucosal break that are continuous between the tops of two or more mucosal folds but involve equal to or less than 75% of the circumference. The image samples are listed in Figure 1. We divided all images into three groups, i.e., LA grade A–B, LA grade C–D, and normal EC-J.

### 2.2. Study Design

In this study, 2000 adult cases were collected retrospectively from the endoscopic system at Chung Shan Medical University Hospital from December 2019 to March 2020. Here, an image quality evaluation was performed to confirm the intactness of the EC-J image, identify the image resolution, and identify any foreign body interference. The endoscopic images of the EC-J from the 496 patients that passed the quality evaluation were then divided into development and test sets. Images of 464 cases were selected as the development set; however, not all cases had both conventional and NBI pictures. Eventually, we obtained 247 GERD A–B images, 155 GERD C–D images, and 62 normal EC-J images from the conventional images. Initially, we obtained 244 GERD A–B images, 157 GERD C–D, images, and 48 normal EC-J images from the NBI images. Note that the original image set was a clinical dataset; thus, data imbalance was evident. In addition, the GERD images had rotation invariance; therefore, this study first employed a static data augmentation approach to overcome image skewness in some categories. Specifically, 222 and 233 images were augmented in the NBI and conventional modes by rotating the axis of the original GERD C–D and normal EC-J images, respectively. Finally, we constructed a balanced development set for AI model training and internal validation. For external validation, we reserved 32 images to test the recognition of the trained AI system and inexperienced trainees. The tests of the young trainees, who were blinded to this study, were performed using an email system. A detailed flowchart of the study design is shown in Figure 2.

### 2.3. Model Development

In this study, the visual geometry group (VGG) neural network model pretrained by ImageNet [24] was employed as the base model for image feature extraction. This technique adjusts the structure of certain pretrained neural network models using transfer learning [25] to perform other different image classification tasks.

In consideration of the balance between network capacity and validation accuracy and by discussing the influence of different regularization and optimization strategies [26], we designed a deep convolutional neural network (CNN) architecture with high generalizability. The proposed CNN architecture is called GERD-VGGNet (Figure 3).

The proposed GERD-VGGNet architecture includes 13 convolutional layers, five max pooling layers, one global average pooling layer, four dense layers, four batch normalization layers [27], four activation layers using the rectified linear unit (ReLU) function [26,28], and the last dense layer with softmax classification.

This model employs the Adam optimizer with a batch size of 64 examples for two-stage optimization training of the entire network architecture using a non-freezing transfer learning method [26]. Here, in the first training stage, the learning rate is set to $10^{-4}$ for network training up to 600 epochs. The second fine-tuning stage uses 400 epochs for network tuning with a smaller learning rate of $10^{-5}$.

To enhance the generalizability of the model, dynamic data augmentation is considered an effective method to train a generally applicable model using a limited amount of training data [26,29]. To make up for the lack of data, a dynamic data augmentation technique is included in the training. After applying image translation and flipping processing, the original images in the training subset of the development set were altered to create more images to allow the model to continue learning. Specifically, for image translation, we used random shifts in a maximum range of 20% of the total width or height of the image. For image flipping, we randomly applied horizontal and vertical flips. Taking NBI endoscopy as an example, we ended up with a total of ((244 + 229 + 198) × 0.9) × 1000 = 603 × 1000 = 603,000 training images after 1000 learning epochs. It is worth noting that data augmentation should be not performed on the validation set and the test set.

The entire training process applied a callback mechanism to store a copy of the network model parameters each time the accuracy of the validation set was improved. After the training algorithm was terminated, the best network model parameters were selected using the early stopping criterion [30] to obtain the model with the lowest validation set error. Here, we expected that the verification task would support improved generalizability.

### 2.4. Model Evaluation

To evaluate the advantages and disadvantages of the proposed model, we applied 10-fold cross-validation for verification, where the development set was randomly divided into 10 subsets by selecting one subset as the validation set and considering the remaining nine subsets as the training set. This experiment was repeated 10 times until each subset was used as a validation set. Finally, the average and standard deviation of the classification results of all 10 experiments were calculated as indicators of the model’s quality.

The GERD images were recombined into three categories of classification problems, i.e., LA grade A–B, LA grade B–C, and normal. In addition, a confusion matrix was used as a model performance evaluation tool; the overall rate of accuracy and rate of accuracy for each category were calculated.

### 2.5. Classifier Performance Comparison

The statistic *Ps* [31], which uses the classical hypothesis testing paradigm to compare the performance of classifier models M1 and M2, is expressed follows:(1)$$P_{s}=\frac{\left|E_{1}-E_{2}\right|}{\sqrt{q\left(1-q\right)\left(\frac{2}{n}\right)}}$$
where, *E*_1_ and *E*_2_ are the error rates for models M1 and M2, respectively, *q* is (*E*_1_ + *E*_2_)/2, and *n* is the number of examples in the test set. If the value of *Ps* ≥ 2, one can be 95% confident that the difference in the test set performance between models M1 and M2 is significant.

The proposed GERD-VGGNet model was developed using the Python programming language (version 3.8), the TensorFlow 2.3 framework, the Pandas library (version 1.2), and the NumPy library (version 1.19). The complete training and testing processes were performed on an Nvidia GTX 1080 Ti (11 GB RAM) with CUDA version 10.2 and cuDNN version 7.

## 3. Results and Analysis

The development and test sets were collected to evaluate the model’s training and performance. These sets were mutually independent, and there was no identical image record data. Table 1 lists the patient numbers and image numbers in the development and test sets.

### 3.1. Analysis of Experimental Results

Different pretrained models have different degrees of image feature extraction ability. In this study, the development set was employed to compare four common pretrained models, i.e., VGG16 [24], ResNet50 [32], ResNet101 [32], and InceptionV3 [33], into the proposed classification model. Classification accuracy was evaluated to determine which model was the most appropriate for use as the pretrained model. The results of the training using the four models are compared in Figure 4. In addition, the time costs of model training were evaluated. Here Resnet 101 required the most time, which consumed more CPU time than inceptionv3, res-net50, and VGG16 (in order of decreasing time cost). The results confirm that the pretrained VGG16 model demonstrated the best training accuracy, validation accuracy, and lowest time costs. Thus, in subsequent testing, the pretrained VGG16 model was used as the image feature extraction model.

Figure 5a shows the model training history in the NBI mode without dynamic data augmentation. As can be seen, the validation accuracy trained by the model shows worse performance; however, with dynamic data augmentation, as shown in Figure 5b, the validation accuracy is better performance, which demonstrates showing that the dynamic data augmentation technique improved classification performance (Figure 6).

In fact, the number of original training images was relatively small, i.e., only 603 samples were used in the no data augmentation case. After 1000 learning epochs, the number of augmented training images was increased to 603,000 samples using dynamic data augmentation. As shown in Figure 6, the experimental results clearly indicate that implementing data augmentation in the training process realized better accuracy for both the training and validation sets than training without data augmentation. The results shown in Figure 6 demonstrate that accuracy was improved from 98.5% to 100% in the training set and 59.5% to 89.3% in the validation set using data augmentation. When data augmentation was used for model training, the trained model reduced the overfitting phenomenon effectively. In addition, data augmentation ensured good model generalizability.

Figure 7 compares the classification performance of the proposed GERD-VGGNet model and four machine learning models, i.e., the RBF-SVM, Decision Tree, Random Forest, and Adaboost classifiers. As can be seen, the training and validation accuracies of the proposed GERD-VGGNet are better than that of the compared classifiers.

### 3.2. Model Training and Validation Performance Evaluation

After conducting 10-fold cross-validation, the training, validation, and overall (mean ± standard deviation) accuracy rates were 1.000 ± 0.001, 0.893 ± 0.050, and 0.989 ± 0.005, respectively, in NBI mode. For the conventional mode, the training, validation, and overall accuracy rates were 1.000 ± 0.001, 0.865 ± 0.042, and 0.986 ± 0.004, respectively. These results demonstrate that the model training and validation performance of NBI mode was slightly better than that of the conventional mode. The confusion matrix of the proposed GERD-VGGNet model for the NBI development set is shown in Table 2. Here, the results demonstrate that the overall accuracy rate was up to 0.997, and two images were misclassified in each of the A–B and C–D categories. In contrast, with the conventional mode, the overall accuracy rate was 0.994 (Table 2), and two images were misclassified in the A–B category, three images were misclassified in the C–D category, and one image was misclassified in the normal category.

### 3.3. Model Testing Performance Evaluation

Table 3 shows the confusion matrix of the proposed model and the results of the trainees. In the NBI case, the results demonstrate that the proposed GERD-VGGNet misclassified two images of GERD A–B grade, one image of GERD C–D grade, and one image of normal. In addition, the overall accuracy rates were 0.875 for GERD-VGGNet, 0.750 for trainee 1, and 0.656 for trainee 2. The accuracy rates for the A–B category were 0.833 for GERD-VGGNet, 0.750 for trainee 1, and 0.417 for trainee 2. The accuracy rates for the C–D category were 1.0 for GERD-VGGNet, 0.7 for trainee 1, and 0.7 for trainee 2. The correct rates for the normal category were 0.8 for GERD-VGGNet, 0.8 for trainee 1, and 0.9 for trainee 2. The overall accuracy under NBI endoscopy appeared to be better in the AI model than it was for the trainees.

For the conventional mode, the results demonstrate that seven imagers were misclassified by the proposed GERD-VGGNet), 10 images were misclassified by trainee 1, and seven images were misclassified by trainee 2. This means that the overall accuracy was 0.781 for GERD-VGGNet, 0.688 for trainee 1, and 0.781 for trainee 2. The accuracy rates of the A–B category were 0.917 for the proposed GERD-VGGNet, 0.667 for trainee 1, and 0.667 for trainee 2. For the C–D category, the accuracy rates were 0.6 for the proposed GERD-VGGNet, 0.7 for trainee 1, and 0.8 for trainee 2. The correct rates for the normal category were 0.8 for the proposed GERD-VGGNet, 0.7 for trainee 1, and 0.9 for trainee 2. The overall accuracy under conventional endoscopy was similar between the AI model and the trainees (Table 3).

Table 4 compares the Ps values of the four classifier models. Here, we conclude that, by using the NBI mode, the proposed GERD-VGGNet outperformed trainee 2 at 95% confidence level, as marked * in Table 4. In addition, the proposed GERD-VGGNet outperformed trainee 1 under NBI endoscopy; however, statistical significance was not observed in this case.

Table 5 compares the performance of the proposed GERD-VGGNet model and the existing methods [22,23]. As shown in Table 5, the proposed model outperformed the method proposed by Huang et al. [23]. In addition, the proposed model demonstrated similar performance as the method proposed by Pace et al. [22]. Note that the existing methods [22,23] were only applicable to binary classification of GERD, and one method [22] used questionnaire data rather than image data. We found that direct use of image data to predict the GERD grade is more beneficial to clinical diagnosis than collecting questionnaire data.

## 4. Discussion

### 4.1. Model Training and Validation Performance

The original image set was a clinical dataset; thus, data imbalance was evident in the dataset. Therefore, we employed data augmentation to overcome the image skewness problem in some categories. We then applied a dynamic data augmentation approach for AI model training. The experimental results clearly demonstrate that data augmentation is key to training a deep learning neural network. The results shown in Figure 5 and Figure 6 clearly demonstrate that the trained model exhibited serious overfitting problems when data augmentation technology was not applied. In contrast, with data augmentation, the established model effectively reduced the overfitting phenomenon and demonstrated good generalizability.

In our analysis, the AI interpretation of GERD endoscopic classifications improved after deep learning, and the validation quality was good, especially after at least 600 learning epochs (Figure 4). The external validation demonstrated that the test accuracy of the proposed GERD-VGGNet model was 91.7% for the conventional A–B grade, 60% for the conventional C–D grade, and 80% for the normal group). In addition, the external validation showed that the test accuracy of the proposed GERD-VGGNet model under NBI endoscopy was 83.3% for the A–B grade, 100% for the C–D grade, and 80% for the normal group. The overall prediction accuracy for normal cases increased under NBI endoscopy compared to conventional endoscopy, and this phenomenon is consistent with previous studies that investigated manual interpretation [13].

Overall, the experimental results indicate that the proposed method can automatically diagnose and grade GERD without manual selection of a region of interest, with automatic feature extraction from image data, and achieve better accuracy compared to state-of-the-art AI systems for endoscopic GERD classification. To the best of our knowledge, this study is the first to develop a deep learning model for computer-aided diagnosis and automatic GERD grading according to LA grades.

### 4.2. Performance of NBI in AI Prediction

The prediction accuracy under NBI was significantly better with the proposed GERD-VGGNet model compared to trainee 2. The proposed GERD-VGGNet model under NBI endoscopy obtained higher accuracy than conventional endoscopy; however, the difference did not demonstrate statistical significance. These results suggest that interpretation of the AI model can be influenced by image contrast, which implies that NBI images can be interpreted with better accuracy. NBI endoscopy enhances the contrast of the mucosal surface and helps diagnose and grade GERD in manual interpretations [34]. This effect is similar to previous NBI applications in GERD [34,35] and in the NBI-guided diagnosis of Barrett’s esophagus in England [36], but it was first confirmed in the proposed AI prediction model.

### 4.3. Limitations

This was a pioneer study in endoscopic GERD LA classification comparisons with trainees; thus, we acknowledge that the number of examined cases was small. In addition, the comparison of conventional endoscopy and NBI endoscopy further limited our case numbers because NBI observation is not performed routinely in our daily practice, particularly in case of normal or grade A GERD under initial conventional endoscopy. Although our test set was small, statistical significance was observed between the proposed GERD-VGGNet model and trainee 2 under the NBI endoscopy. Thus, due to the limited amount of data, future large-scale studies are required to further confirm the results presented in this paper.

## 5. Conclusions

In this paper, we have proposed the GERD-VGGNet model. The experimental results have confirmed that the proposed model and training strategies can automatically diagnose and grade GERD without manual selection of a region of interest and achieves higher accuracy than state-of-the-art AI systems. Given the outcomes of the interpretation of LA GERD classification in the AI model, we believe that the proposed GERD-VGGNet model can assist endoscopic findings for trainees and that NBI endoscopy increases the accuracy of the interpretation results in AI systems, which has been previously demonstrated in manual interpretations.

In future research, we will try to integrate different XAI (explainable artificial intelligence) analysis technologies and attention models to improve interpretation capabilities of the proposed AI model.

## Figures and Tables

**Figure 1 ijerph-18-02428-f001:**
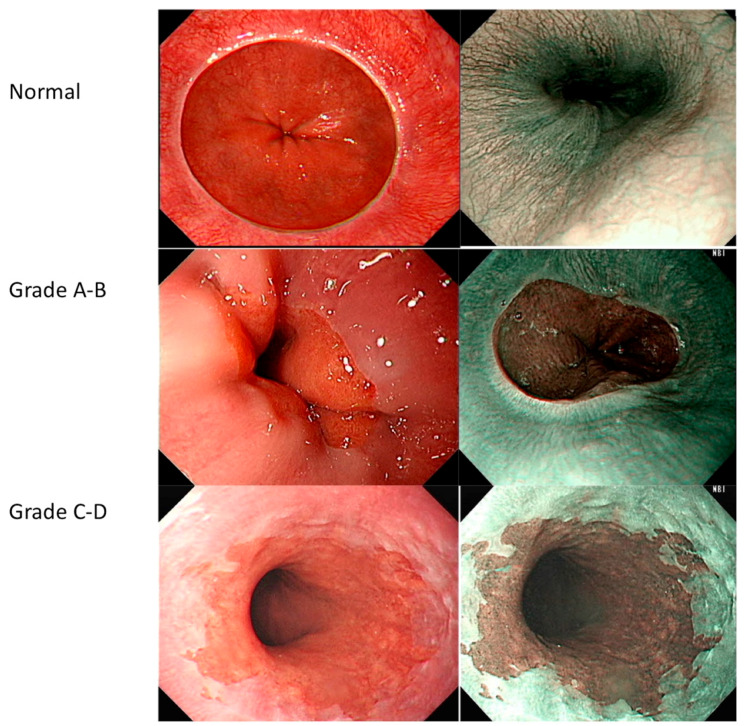
Sample images to illustrate Los Angeles classification of gastroesophageal reflux disease.

**Figure 2 ijerph-18-02428-f002:**
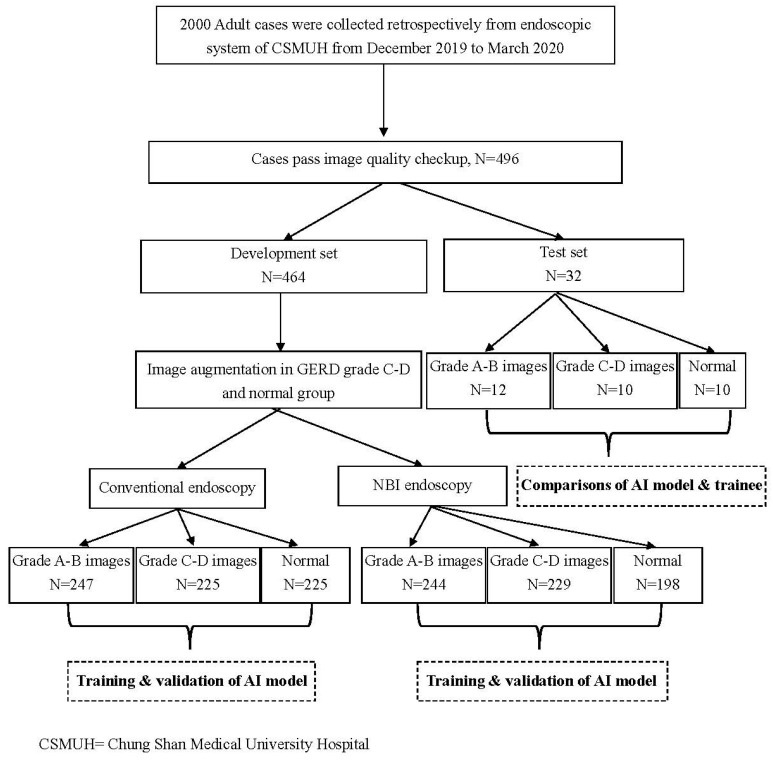
Flowchart of the study design.

**Figure 3 ijerph-18-02428-f003:**
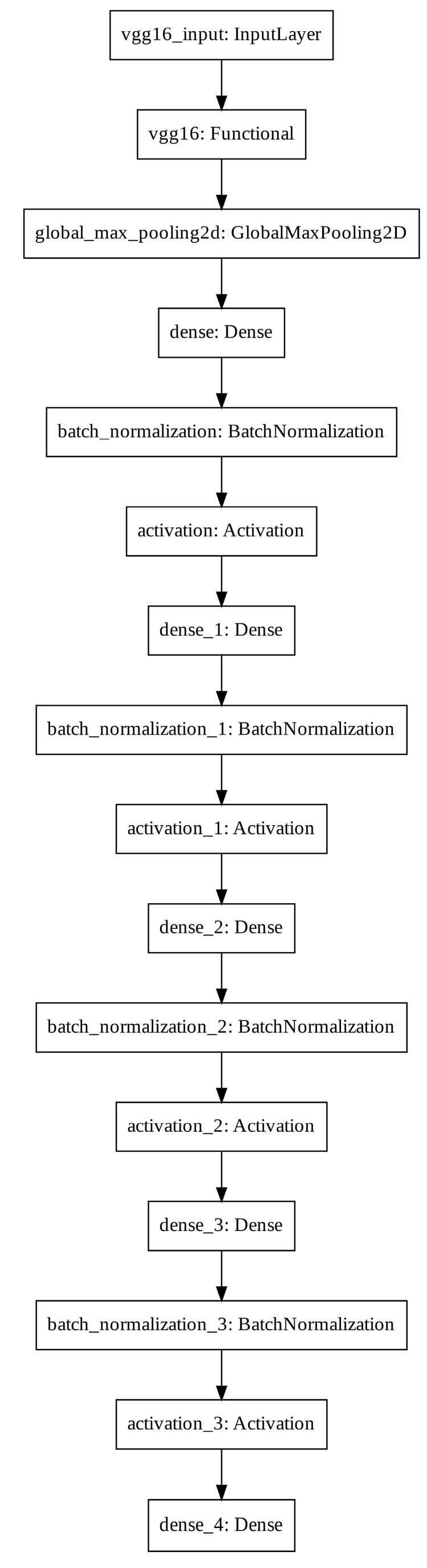
Convolutional neural network classifier architecture of the proposed GERD-VGGNet model.

**Figure 4 ijerph-18-02428-f004:**
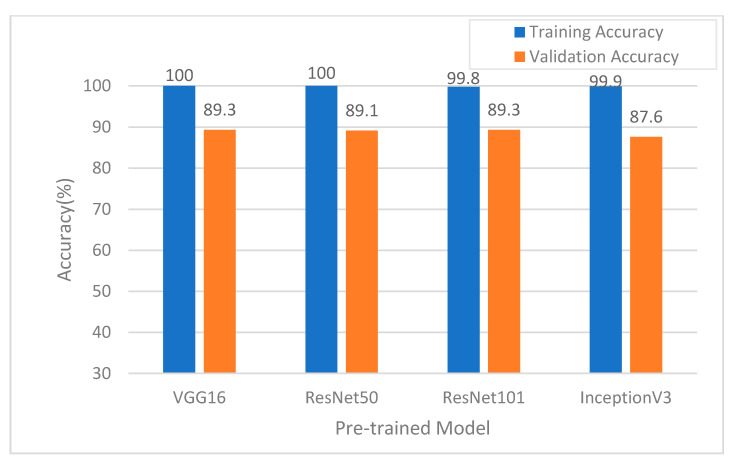
Comparison of accuracy using different pretrained models in the narrow-band image (NBI) mode.

**Figure 5 ijerph-18-02428-f005:**
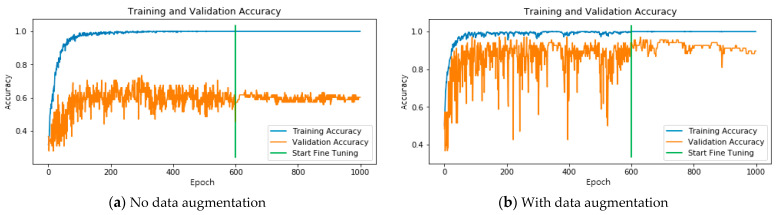
Model training history in the NBI mode: (**a**) without data augmentation; and (**b**) with data augmentation.

**Figure 6 ijerph-18-02428-f006:**
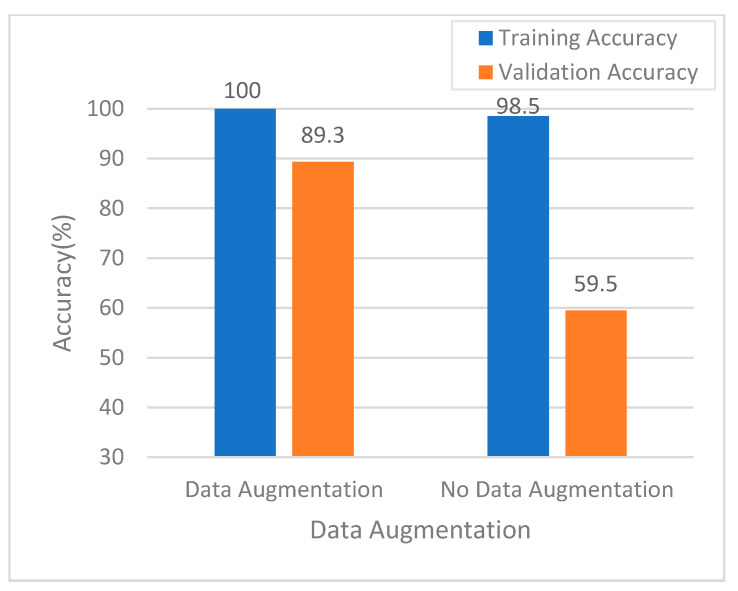
Comparison of accuracy using data augmentation in the NBI mode.

**Figure 7 ijerph-18-02428-f007:**
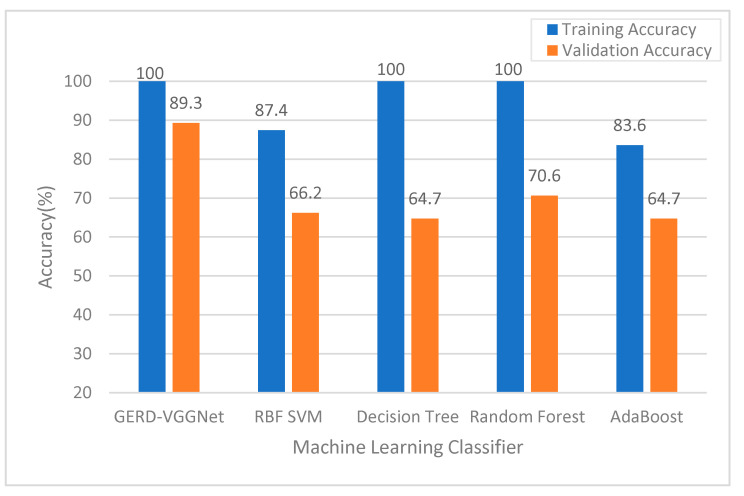
Comparison of accuracy using different classifiers in the NBI mode.

**Table 1 ijerph-18-02428-t001:** Baseline characteristics of development and test sets.

Characteristics Patient Number	Development Set N = 464		Test Set N = 32	
	N	%	N	%
Conventional images				
LA grade ^1^ A–B	247	35.4	12	37.5
LA grade C–D	225	32.3	10	31.3
Normal	225	32.3	10	31.3
NBI ^2^ images				
LA grade A–B	244	36.3	12	37.5
LA grade C–D	229	34.2	10	31.3
Normal	198	29.5	10	31.3
Augmentation (conventional)				
LA grade A–B	0	NA	0	NA
LA grade C–D	70	NA	0	NA
Normal	163	NA	0	NA
Augmentation (NBI)				
LA grade A–B	0	NA	0	NA
LA grade C–D	72	NA	0	NA
Normal	150	NA	0	NA

^1^ Los Angeles classification = LA grade. ^2^ Narrow-band image = NBI.

**Table 2 ijerph-18-02428-t002:** Confusion matrix of proposed GERD-VGGNet on the development set.

Image Type		Conventional			NBI ^2^		
Real LA ^1^ Classification		A–B	C–D	Normal	A–B	C–D	Normal
**GERD-VGGNet**	A–B	247	4	0	242	0	0
	C–D	0	221	0	1	229	0
	Normal	0	0	225	1	0	198
	Accuracy	100%	98.2%	100%	99.2%	100%	100%

^1^ Los Angeles classification = LA grade. ^2^ Narrow-band image = NBI.

**Table 3 ijerph-18-02428-t003:** Confusion matrices of GERD-VGGNet and trainees on the test set.

Image Type		Conventional				NBI ^2^		
Real LA ^1^ Classification		A–B	C–D	Normal	A–B		C–D	Normal
**GERD-VGGNet**	A–B	11	4	2	10		0	2
	C–D	1	6	0	1		10	0
	Normal	0	0	8	1		0	8
	Accuracy	91.7%	60%	80%	83.3%		100%	80%
**Trainee 1**	A–B	8	3	2	9		3	2
	C–D	2	7	1	1		7	0
	Normal	2	0	7	2		0	8
	Accuracy	66.7%	70%	70%	75%		70%	80%
**Trainee 2**	A–B	8	2	1	5		3	1
	C–D	3	8	0	5		7	0
	Normal	1	0	9	2		0	9
	Accuracy	66.7%	80%	90%	41.7%		70%	90%

^1^ Los Angeles classification = LA grade. ^2^ Narrow-band image = NBI.

**Table 4 ijerph-18-02428-t004:** Model comparisons between the proposed AI model and trainees with NBI and conventional endoscopy.

Model 1	Model 2	Ps
GERD-VGGNet-NBI ^1^	Trainee1-NBI	1.281
GERD-VGGNet-NBI	Trainee 2-NBI	2.068 *
Trainee 1-NBI	Trainee 2-NBI	0.823
GERD-VGGNet-conventional	Trainee 1-conventional	0.552
GERD-VGGNet-conventional	Trainee 2-conventional	0.293
Trainee 1-conventional	Trainee 2-conventional	0.842
GERD-VGGNet-NBI	GERD-VGGNet -conventional	1.281
Trainee 1-NBI	Trainee 1-conventional	0.552
Trainee 2-NBI	Trainee 2-conventional	1.112

^1^ Narrow-band image = NBI.

**Table 5 ijerph-18-02428-t005:** Performance comparison of different AI systems for prediction of gastroesophageal reflux disease.

Task	Algorithm	Data Used	Evaluation Method	Overall Accuracy	Sensitivity	Specificity
Binary classification	Machine learning (ANN) [22]	QUID ^1^ questionnaire (577 GERD ^2^ patients, 94 normal cases)	hold-out	99.2%	99.1%	99.8%
Binary classification	Machine learning (HHDF-SVM) [23]	147 RGB images (39 GERD patients, 108 normal cases)	10-fold cross-validation	93.2%	94.9%	92.6%
Three-class classification	Deep learning + data augmentation (proposed GERD-VGGNet)	603,068 NBI ^3^ images (GERD A–B: GERD C–D: normal EC-J = 244:229:198)	10-fold cross validation	98.9% ± 1%	99.8% ± 0.2%	99.7%± 0.2%

^1^ QUestionario Italiano Diagnostico = QUID. ^2^ Gastroesophageal reflux disease = GERD. ^3^ Narrow-band image = NBI.

## Data Availability

All the data of images and analysis process were kept at the lab of M.-H.T.

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
