# Peer review of "A Deep Learning Model for Classification of Endoscopic Gastroesophageal Reflux Disease"

_ijerph, 2021, doi:10.3390/ijerph18052428_

Round 1
Reviewer 1 Report
This manuscript introduces the work on classifying the images of gastroesophageal reflux disease using deep learning. This is an interesting topic. The manuscript presents the performance by comparing it with the results delivered by humans (two trainees).
By carefully reviewing the manuscript, the following issues are identified and need to be addressed.
1. Issues in Abstract
- The motivation is not clearly presented
- As far as I can understand, the neural network (CNN) is the main component in developing this model. However, it is not mentioned in the abstract.
- The abstract presents the results that "In the test set, the AI model had an accuracy of 87.5%, which was significantly better than the results of trainees (75.0% & 65.6%).". However, this is just part (incomplete) results. Using incomplete results to represent the entire results causes misleading.
- A clear conclusion is missing in the abstract.
2. Issues in Introduction
- The motivation/rationale is not well introduced. So that the scientific question(s) is unclear. For example, why was deep learning adopted? What is the benefit brought by deep learning?
- What does "conventional endoscopy" mean?
3. Issues in Methods
- Line 80: "We compared the interpretation of the endoscopic pictures under conventional and NBI endoscopy by AI in the first step." What does the "compare" refer to? how were they compared?
- Figure 2 shows that 2000 cases were collected. However, quality checkup is not reflected in the description of the methods.
- Figure 2 is not a study design, but the baseline characters. how the dataset (images) were constructed.
- Line 89: 498, however, Line 108: 496. Which should be the correct amount of images?
- Line 120: How were the 32 images selected for the test? Why 32? The number of test images (n = 32) is low.
- The number of trainees (n = 2) is low too.
- What is the information about them, what is their expertise? What if not the trainees but more experienced persons?
- Line 138: Equation (1) is not shown in a proper way.
- I failed to understand fig 3, what do these terms in the boxes mean? what do the numbers in () mean?
- Many methods, without any explanation on their working principle and the motivation of using them. E.g., non-freezing transfer learning method, 2-stage optimization training, dynamic data augmentation, translating and flipping methods, etc.
4. Issues in Results
- Figure 4: the unit/labels of the x-/y- axis are missing, making it hard to understand.
- Line 198: Not just for "performance evaluation", but also for model training.
- Table 3: some calculation (Accuracy of Trainee 2) is wrong
- Line 267 -- 269: The statement belongs to the discussion; the results cannot support this statement.
- Table 4 also shows the comparison of the performance between different models, even trainees. However, a question can be raised up: What are the motivation and the purposes of the comparing?
- Table 5: 603,068 --> 603 and 68
- In Table 5, the performance of the proposed model was compared with the other machine learning algorithms. However, based on the results, it is hard to conclude the model in this work is significantly better than the other. They are close.
5. Issues in Discussion
- Line 271 -- 278: I cannot see the topic sentence of this paragraph. Besides, repeating methods is not necessary for discussion.
- Line 285 -- 287: What could be the reasons to explain that?
6. Issues in Conclusion
- The conclusion is questionable with just 32 test images and comparing the performance of the model with just two trainees.
- Line 316 -- 317: I failed to understand this statement.
7. Others
- With respect to the language, the manuscript needs to be extensively improved for publication.
- The definitions of many abbreviations in the whole manuscript are missing. This causes the content, particularly the methods, hard to follow.
- The full term of NBI is shown in every figure that the term appears, I don't think it is necessary. However, the other abbreviations were not present in such away.
Reviewer 2 Report
- The original dataset is too small
- It is a good idea to compare the learning results of the model with real clinical data. This kind of research can be used as a reference model for practical use
- In this study, they end up with 603 x 1000 = 603,000 training images by considering that the augmentation of the training set is dynamic. The original data volume was only 603. After the data was increased, it reached 603,000. However, there may increase more overfitting problems. Therefore, the details and reasons why the multiple expansion of this data should be explained and why at the beginning, the accuracy of both the training set and the test set was very low, only about 46.9, but after the data augmentation, the accuracy of the training set reached to 99.4, is this due to the model overfitting?
- The data in the original dataset is too small ,so this research choose Data Augmentations to increase the data is a good idea. But I suggest that for the data and the results after Data Augmentation, this might have more discussion to state out the differences before and after this data augmentation.

Reviewer 3 Report
In this study (Development of a Deep Learning Model for Endoscopic Gas-troesophageal Reflux Disease Classification), the authors proposed a novel deep learning model, GERD-VGGNet, for Gastroesophageal reflux disease (GERD) classification. By using the data augmentation method, the proposed method achieved an overall accuracy > 98% with 10-fold cross-validation. The results were much higher than the prediction results of trainees. Overall, the manuscript is well written and decently organized. However, no other deep learning baseline methods were provided in the given study. Without comparing other commonly used deep learning-based methods, it is hard to believe the proposed method could show to be higher generalization compare among the other widely used deep learning models ( e.g: ResNet-101 [1] or Inception network [2]).
Specific comments:
Q1: Discuss the importance of feature: Based on the proposed method, what kinds of features are more discriminative for certain GERD classes? It is hard to interpret what kinds of features are more related to certain GERD for classification when using the proposed method. I suggested that the authors could consider adding attention models [3] for feature visualization.
Q2: To obtain more interpretable models like the study in [4], I suggested that the authors could consider using traditional feature selection methods with machine learning classifiers, such as random forest or SVM, for classification.
Q3: Also compare one or two commonly used deep learning models, such as ResNet-101 and Inception network for classification.
Q4: The description of how to design and optimized the deep learning architecture is not clear to me. Is the proposed GERD-VGGNet architecture was optimized based on a given training dataset?
[1] He, Kaiming, et al. "Deep residual learning for image recognition." Proceedings of the IEEE conference on computer vision and pattern recognition. 2016.
[2] Szegedy, Christian, et al. "Rethinking the inception architecture for computer vision." Proceedings of the IEEE conference on computer vision and pattern recognition. 2016.
[3] Zhang, Zizhao, et al. "Mdnet: A semantically and visually interpretable medical image diagnosis network." Proceedings of the IEEE conference on computer vision and pattern recognition. 2017.
[4] Rosenfeld, Avi, et al. "Development and validation of a risk prediction model to diagnose Barrett's oesophagus (MARK-BE): a case-control machine learning approach." The Lancet Digital Health 2.1 (2020): e37-e48.
Round 2
Reviewer 1 Report
Thanks for considering my comments. I am fine with the current revision.
Reviewer 3 Report
The authors have addressed all of my comments.